# YPL-001 Shows Various Beneficial Effects against Cigarette Smoke Extract-Induced Emphysema Formation: Anti-Inflammatory, Anti-Oxidative, and Anti-Apoptotic Effects

**DOI:** 10.3390/antiox12010015

**Published:** 2022-12-22

**Authors:** Kyoung-Hee Lee, Jisu Woo, Jiyeon Kim, Chang-Hoon Lee, Chul-Gyu Yoo

**Affiliations:** 1Division of Pulmonary and Critical Care Medicine, Department of Internal Medicine, Seoul National University Hospital, Seoul 03080, Republic of Korea; 2Department of Internal Medicine, Seoul National University College of Medicine, Seoul 03080, Republic of Korea

**Keywords:** YPL-001, cigarette smoke extracts, IL8, NF-κB, HDAC, Nrf2, apoptosis

## Abstract

Inflammation, oxidative stress, and apoptosis are thought to be important causes of chronic obstructive pulmonary disease (COPD). We investigated the effect of YPL-001 (under phase 2a study, ClinicalTrials.gov identifier NCT02272634), a drug derived from *Pseudolysimachion rotundum* var. *subintegrum*, on cigarette smoke extract (CSE)-induced inflammation, the anti-oxidative pathway, and apoptosis in human lung epithelial cells and on CSE-induced emphysema in mice. YPL-001 suppressed CSE-induced expression of IL8 mRNA and protein. This was due to the reduction in NF-κB transcriptional activity by YPL-001, which resulted from the blockade of acetylation of the NF-κB subunit p65 (Lys310). Histone deacetylases (HDACs) prevent gene transcription by condensing the DNA structure and affecting NF-κB nuclear binding. YPL-001 alone increased HDAC2 activity and enhanced CSE-induced activation of HDAC2. YPL-001-induced suppression of NF-κB transcriptional activity might be caused by increased HDAC2 activity. YPL-001 increased nuclear factor (erythroid-derived 2)-like 2 (Nrf2) expression via both degradation of its inhibitory protein, Kelch-like ECH-associated protein 1, and an increase in de novo protein synthesis. YPL-001 increased the DNA binding activity of Nrf2. Consequently, YPL-001 upregulated the expression of Nrf2-targeted anti-oxidant genes such as NAD(P)H quinone dehydrogenase 1 and heme oxygenase 1. Moreover, YPL-001 significantly suppressed CSE-induced apoptotic cell death. In vivo study showed that CSE-induced emphysematous changes, neutrophilic inflammation, protein leakage into bronchoalveolar space, and lung cell apoptosis in mice were suppressed by YPL-001 treatment. Taken together, these results suggest that YPL-001 is a good therapeutic candidate for the treatment of COPD by blocking inflammation and apoptosis and activating the anti-oxidative pathway.

## 1. Introduction

Chronic obstructive pulmonary disease (COPD) is a type of lung disease that is characterized by airflow limitation resulting from airway inflammation, remodeling, and parenchymal destruction. It is one of the leading causes of death worldwide and its prevalence is increasing [1,2]. Treatments involving the combined use of various bronchodilators and corticosteroids help slow the progression of COPD and control its symptoms. However, there is no cure for COPD. Therefore, finding effective therapeutic alternatives for treating COPD is imperative.

Both neutrophilic inflammation and increased levels of interleukin-8 (IL8) are prominent features of COPD [3,4,5]. IL8 is a neutrophil chemotactic factor that is produced by several types of cell. It is well known that cigarette smoke extract (CSE) induces IL8 production in bronchial epithelial cells [6]. Exposure to cigarette smoke (CS) or CSE activates numerous intracellular signaling pathways including the nuclear factor-kappa B (NF-κB) pathway [7]. NF-κB, a pleotropic transcription factor, is normally sequestered in the cytoplasm in an inactive form, bound to its inhibitory protein, IκB. NF-κB promotes the expression of various cytokines such as IL8 and TNF-α. IκB kinase (IKK) is activated upon cell stimulation. Active IKK phosphorylates IκBα at serine 32/36 residues, leading to IκBα degradation via a proteasome. As a result, NF-κB is translocated into the nucleus, where it binds to the κB element and transactivates the downstream genes [8,9,10].

Nuclear factor (erythroid-derived 2)-like 2 (NFE2L2) is a gene that encodes for the transcription factor Nrf2 that is expressed predominantly in the epithelium and alveolar macrophages. It regulates the expression of anti-oxidants and detoxification genes, and is known to protect the lungs from oxidative airway diseases [11]. Under normal or unstressed conditions, cytosolic Nrf2 binds to its cytosolic inhibitor Kelch-like ECH-associated protein 1 (KEAP1). KEAP1 is an adaptor for cullin-3 E3 ubiquitin ligase, and the binding of KEAP1 to Nrf2 mediates ubiquitination and the subsequent degradation of Nrf2. In response to various stresses including oxidative stress, KEAP1 is inactivated and Nrf2 becomes phosphorylated. Phosphorylated Nrf2 accumulates in the nucleus. Nrf2 forms a complex with other transcription factors and cofactors, and binds to the anti-oxidant response element (ARE), leading to upregulation of ARE-dependent genes such as NAD(P)H quinone dehydrogenase 1 (*NQO1*) and heme oxygenase 1 (*HMOX1*). Previous studies have shown that deficiency of Nrf2 or HMOX1 increases susceptibility to CS-induced emphysema, and increases lung cell apoptosis and inflammation [12,13]. Therefore, Nrf2 targeting might be a good therapeutic strategy for COPD treatment.

*Pseudolysimachion rotundum* var. *subintegrum* (Plantaginaceae) is an herb with a long lifespan and is found in Korea and China [14]; it has been used as a traditional herbal medicine to treat various respiratory diseases. YPL-001 is methanol extract of *Pseudolysimachion rotundum* var. *subintegrum,* and is under investigation in a Phase 2a clinical trial in COPD patients (ClinicalTrials.gov identifier NCT02272634). Verproside, an important active ingredient of YPL-001, and seven other compounds were identified through high-performance liquid chromatography (HPLC), thin layer chromatography (TLC) fingerprint, nuclear magnetic resonance spectroscopy (NMR) spectrum, and mass spectrum analysis (Table 1). However, the effects of YPL-001 on CSE-induced emphysema are not known, and its regulatory mechanisms are unclear.

In this study, we demonstrated that YPL-001 suppresses CSE-induced emphysema formation via multiple mechanisms such as anti-inflammatory, anti-oxidative, and anti-apoptotic effects. These results suggest that YPL-001 is a promising therapeutic candidate for the treatment of COPD.

## 2. Materials and Methods

### 2.1. Cells and Reagents

BEAS-2B, normal human bronchial epithelial cells, and normal primary human bronchial epithelial cells (HBECs) were maintained in defined keratinocyte serum-free medium (GIBCO by Life Technologies, Grand Island, NY, USA). Normal primary HBECs were collected after review and approval by the Seoul National University Hospital Institutional Review Board (SNUH IRB number: H-1602-108-742). YPL-001 (the drug substance derived from *P. rotundum* var. *subintegrum*) was produced by Yungjin Pharmaceutical Co., Ltd. (Seoul, Republic of Korea) according to the process described in the International Conference on Harmonization (ICH) and Food and Drug Administration (FDA) guidelines (Korean patent 10-1476095). Anti-acetyl-NF-κB p65 (Lys310), anti-HDAC2, anti-Nrf2, anti-KEAP1, anti-caspase-3, and anti-NQO1 antibodies were purchased from Cell Signaling (Danvers, MA, USA). Anti-poly (ADP-ribose) polymerase (PARP), anti-p65, anti-HMOX1, and anti-GAPDH antibodies were from Santa Cruz Biotechnology (Santa Cruz, CA, USA).

### 2.2. CSE Preparation

CSE was prepared as described in other studies [15]. Commercial cigarettes (THIS; purchased from Korea Tomorrow & Global Co., Ltd., Daejeon, Republic of Korea) were smoked via a bottle system connected to a vacuum pump. The smoke from 20 cigarettes was bubbled through 60 mL of phosphate-buffered saline (PBS) (GIBCO). The large insoluble particles were removed using a 0.22 μm filter.

### 2.3. Determination of Cytokine Secretion

The levels of cytokine in the culture supernatants were determined using a Bio-Plex Pro™ cytokine assay kit (Bio-Rad, Hercules, CA, USA).

### 2.4. Quantitative Real-Time PCR

Quantitative real-time PCR analysis was performed as described in a previous study [6]. An Nrf2 probe (Hs00975961_g1), an NQO1 probe (Hs00168547_m1), and a GAPDH probe (Hs99999905_m1) were obtained from Applied Biosystems (Foster City, CA, USA). We used the following primers: human IL8 (CXCL8, fwd: 5′-GCAGCTCTGTGTGAA GGTGC-3′, rev: 5′-TCTGCACCCAGTTTTCCTTG-3′); heme oxygenase 1 (HMOX1, fwd: 5′-CAGGCAGAGAATGCTGAGTTC-3′, rev: 5′-GCTTCACATAGCGCTGCA-3′); human GAPDH (fwd: 5′-GAAGGTGAAGGTCGGAGTC-3′, rev: 5′-GAAGATGGTGATGG GATTTC-3′); mouse keratinocyte chemoattractant (KC) (fwd: 5′-TGTCAGTGCCTGCAG ACCAT-3′, rev: 5′-CCTGAGGGCAACACCTTCA-3′); mouse macrophage inflammatory protein-2 (MIP2) (fwd: 5′-CCAACCACCAGGCTACAGG-3′, rev: 5′-GCGTCACACTCAA GCTCTG-3′); mouse GAPDH (fwd: 5′-ACGGCAAATTCAACGGCACAG-3′, rev: 5′-TGG GGGCATCGGCAGAAGG-3′).

### 2.5. NF-κB p65 Immunofluorescent Staining

Cells were fixed, permeabilized, and p65 immunofluorescent staining was performed as previously described [16].

### 2.6. NF-κB Luciferase Activity Assay

Cells cultured in 35 mm dishes were transfected with the NF-κB reporter plasmid or the control plasmid using a Neon™ transfection system (Invitrogen, Carlsbad, CA, USA). Cell lysates were analyzed using the Luciferase Reporter Assay System (Promega, Madison, WI, USA) according to the manufacturer’s instructions.

### 2.7. Protein Extraction and Western Blot Analysis

Total cell lysates were prepared in 1× cell lysis buffer (Cell Signaling). Nuclear/cytoplasmic proteins were extracted as described in previous studies [15,16]. Protein concentrations were measured using the Bradford protein assay (Bio-Rad). Western blot analysis was performed as previously described [15,16]. The membranes were developed using a SuperSignal West Pico Chemiluminescent kit (Thermo Fisher Scientific, Waltham, MA, USA).

### 2.8. HDAC2 Activity Assay

The HDAC2 activity in the nuclear extracts was determined using a colorimetric HDAC assay kit (Millipore, Temecula, CA, USA). Briefly, the nuclear extracts (20 μg) were mixed with 2× assay buffer and 4 mM substrate in a 96 well plate. The plate was incubated at 37 °C for 1 h. Activator solution was added to each well and the plate was incubated at room temperature for 20 min. The absorbance was read at 405 nm.

### 2.9. Nrf2 Activity Assay

The Nrf2 activity in the nuclear protein was determined using a TransAM^®^ Nrf2 activity assay kit (Active Motif, Carlsbad, CA, USA) according to the manufacturer’s instructions. The nuclear extracts were incubated in a 96-well plate, in which an oligonucleotide containing the anti-oxidant response element (ARE) consensus binding site (5′-GTCACA GTGACTCAGCAGAATCTG-3′) was placed. The primary antibody recognized an epitope on the Nrf2 protein upon DNA binding. A secondary antibody was added to each well and a colorimetric reaction was performed. Absorbance (at 450 nm) was read [15].

### 2.10. Lactate Dehydrogenase (LDH) Release Assay

Cytotoxicity was measured by LDH release assay. Released LDH was determined using the CytoTox-ONE^TM^ homogeneous membrane integrity assay kit (Promega, Madison, WI, USA).

### 2.11. Mouse Model

Healthy female C57BL/6 mice (6 weeks old; body weight, 18–20 g) were obtained from Koatech Technology Corporation (Pyeongtaek, Korea). Animal experiments were approved by the Institutional Animal Care and Use Committee (number 17-0144-C1A0) of Seoul National University Hospital, Seoul, Korea. Mice were anaesthetized and instilled intratracheally with saline or CSE (100 μL). CSE was instilled once a week for 8 weeks (saline n = 6, CSE n = 5, YPL-001 + CSE n = 5). The mice were treated orally with YPL-001 (0.5 mg/20g mouse) once daily for 8 weeks. To collect lungs and bronchoalveolar lavage fluid (BALF), the mice were sacrificed on day 1 after the last CSE instillation.

### 2.12. Measurement of Emphysema

Lung tissues were fixed with 4% paraformaldehyde solution. Fixed lung was dehydrated, embedded, sectioned, and stained with hematoxylin and eosin (H&E). Emphysema was quantified by measuring the mean linear intercept (MLI) as described in a previous study [6]. Four randomly selected ×100 fields per specimen were photographed in a blinded manner. The MLI was measured by placing four 1000 μm lines over each field. The total length of each line divided by the number of alveolar intercepts gives the average distance. The non-parenchymal area was not included.

### 2.13. Analysis of BALF

The lungs of mice were lavaged with 1mL of cold PBS. BALF was centrifuged at 2000 rpm at 4 °C for 10 min. The supernatants were collected to measure the levels of protein. Cell distribution in BAL was quantified in cytospin preparations after Diff-Quik dye staining (Sysmex, Kobe, Japan).

### 2.14. TUNEL Assay

Terminal deoxynucleotidyl transferase (TdT) dUTP nick-end labeling (TUNEL) assay was performed to detect apoptotic cells using the ApopTag^®^ Peroxidase In Situ Apotosis detection kit (Merch, Darmstadt, Germany) according to the manufacturer’s instructions.

### 2.15. Statistical Analysis

Data were analyzed using GraphPad software (San Diego, CA, USA). Data were subjected to a two-tailed unpaired *t*-test for analysis of statistical significance. Data from in vitro cell experiments are expressed as the mean ± SD. Data from the experiments using mice are expressed as the mean ± SE. A *p*-value of <0.05 was considered to be significant.

## 3. Results

### 3.1. YPL-001 Suppresses CSE-Induced IL8 Production

IL8, a neutrophil chemotactic factor, is found at elevated levels in the BALF from patients with COPD [17], and CSE induces IL8 release in lung epithelial cells [6]. Therefore, we first evaluated the effect of YPL-001 on CSE-induced IL8 production. As observed in previous studies, CSE increased IL8 production. YPL-001 pretreatment significantly reduced CSE-induced mRNA expression and release of IL8 in BEAS-2B cells and primary HBECs (Figure 1A–C). We next determined whether the inhibitory effect of YPL-001 on IL8 production is stimulus-specific. As IL17A is upregulated in the lung tissue and sputum of patients with COPD, and plays an important pro-inflammatory role in COPD [18,19], we treated cells with IL17A and determined whether YPL-001 also inhibits IL17A-induced IL8 production. YPL-001 pretreatment significantly suppressed IL8 production in the IL17A-treated cells (data not shown), which indicates that the inhibitory effect of YPL-001 on IL8 production is not specific to CSE stimulation.

### 3.2. YPL-001 Suppresses Transcriptional Activity of NF-κB

NF-κB, which is a transcription factor, promotes the expression of various cytokines. NF-κB is normally sequestered in the cytoplasm in an inactive form, bound to its inhibitory protein IκB. Upon cell stimulation, IκBα is degraded by a proteasome, resulting in nuclear translocation of the NF-κB subunit complex [20]. Therefore, we first determined if YPL-001 blocks nuclear translocation of the NF-κB subunit p65 upon CSE stimulation. We examined the subcellular localization of p65 by immunofluorescent staining for cytoplasmic/nuclear p65. Although the majority of p65 was located in the cytoplasm of the control cells, stimulation with CSE led to an increase in p65 protein in the nucleus, which was not suppressed by YPL-001 pretreatment (Figure 2A). These data indicate that the anti-inflammatory effect of YPL-001 is not due to blocking of the nuclear translocation of NF-κB. Next, we determined whether YPL-001 affects NF-κB transcriptional activity in the nucleus. The cells were transiently co-transfected with the NF-κB–luciferase reporter construct and the control plasmid. Twenty-four hours after transfection, the cells were pretreated with YPL-001 and then stimulated with CSE. CSE increased the transcriptional activity of NF-κB. YPL-001 pretreatment significantly suppressed the NF-κB transcriptional activity induced by CSE (Figure 2B). We also observed that although YPL-001 pretreatment did not block the nuclear translocation of NF-κB by TNF-α stimulation, it suppressed transcriptional activity of NF-κB in TNF-α-treated cells (data not shown). These results suggest that YPL-001 negatively regulates the transcriptional activity of NF-κB in the nucleus.

### 3.3. YPL-001 Suppresses CSE-Induced Acetylation of NF-κB via Increase in HDAC2 Activity

It has been reported that the acetylation of p65 at lysine 310 is required for full transcriptional activity of NF-κB [21]. CSE induced the acetylation of p65 (Lys310) 6 h after treatment, and acetylation returned to the basal level after 13 h. CSE-induced acetylation was completely abrogated by YPL-001 pretreatment (Figure 3A). HDACs remove acetyl moieties from the lysine residues of histones, causing rewinding of the DNA, which silences gene transcription [22]. Moreover, HDACs directly affect NF-κB nuclear binding [23,24]. Therefore, we determined whether the suppression of NF-κB activity by YPL-001 is related to changes in HDAC activity. Cells were pretreated with YPL-001 and stimulated with CSE. HDAC2 protein expression and activity were determined using nuclear extracts. We found that YPL-001 alone increased HDAC2 protein expression and HDAC2 activity. YPL-001 pretreatment enhanced CSE-induced HDAC2 expression and HDAC2 activity (Figure 3B,C). These results suggest that YPL-001 decreases acetylation of NF-κB, which might be mediated by increased HDAC2 activity.

### 3.4. YPL-001 Activates Nrf2

Next, we investigated the effect of YPL-001 on the anti-oxidative pathway. Nrf2 is a major transcription factor that regulates the expression of several anti-oxidant proteins [11]. To determine the effect of YPL-001 on the Nrf2 pathway, we first investigated the effect of YPL-001 on Nrf2 mRNA expression and the nuclear translocation of Nrf2. We treated cells with YPL-001 and extracted the cytoplasmic/nuclear proteins. We used Western blotting analysis to examine the subcellular localization of Nrf2. YPL-001 treatment increased the expression level of Nrf2 mRNA (Figure 4A) and led to the nuclear translocation of Nrf2 and degraded its cytoplasmic/inhibitory protein, KEAP1, in both BEAS-2B cells and primary HBECs (Figure 4B,C). These results suggest that Nrf2 upregulation by YPL-001 is mediated by both KEAP1 degradation and de novo protein synthesis. Nrf2 is known to bind to the ARE and serve as an important regulator of cellular defense against various toxic oxidants. We determined the DNA binding activity of Nrf2. We carried out an Nrf2 activity assay using nuclear proteins. YPL-001 (50 μg/mL) activated Nrf2, and the activity returned to the basal level 12 h after stimulation (Figure 4D). Nrf2 activity induced by a high concentration of YPL-001 (100 μg/mL) was sustained for up to 24 h after stimulation (data not shown). Next, we investigated the effect of YPL-001 on the expression of anti-oxidants such as NQO1 and HMOX1. YPL-001 treatment increased the levels of NQO1 and HMOX1 mRNA expression (Figure 4E,F). The protein expression levels of NQO1 and HMOX1 were upregulated by YPL-001 in both BEAS-2B cells and primary HBECs (Figure 4G,H). Taken together, these data suggest that YPL-001 activates Nrf2.

### 3.5. YPL-001 Suppresses CSE-Induced Apoptotic Cell Death

Not just inflammation and oxidative stress, increased apoptosis of airway epithelial cells is also considered as an important pathogenic factor of COPD [25]. Therefore, we examined the effect of YPL-001 on CSE-induced cell death and apoptosis. Since CSE at a concentration of 1% or less did not affect cell viability, we used 4% CSE to induce epithelial cell death. Cell viability was measured by LDH release assay. YPL-001 pretreatment significantly suppressed CSE-induced cell death in both BEAS-2B cells and primary HBECs (Figure 5A,B). Western blot analyses for PARP cleavage and active caspase-3 were performed to examine if the reduction in cell viability was due to apoptosis. CSE-induced PARP proteolysis and caspase-3 activation were suppressed by YLP-001 pretreatment in a dose-dependent manner (Figure 5C). These results indicate that YPL-001 has a protective effect against CSE-induced apoptotic cell death.

### 3.6. YPL-001 Reduced CSE-Induced Emphysematous Formation via Multiple Mechanisms Such as Anti-Inflammatory and Anti-Apoptotic Effects

To evaluate whether YPL-001 treatment affects the development of CSE-induced emphysema in mice, we treated healthy C57BL/6 mice with CSE once a week for 8 weeks intratracheally to induce emphysema and YPL-001 was administered orally, once a day from day 1 onward (Figure 6A). Alveolar destruction and airspace enlargement were observed in CSE-treated mice, which were suppressed by YPL-001 treatment (Figure 6B,C). CSE treatment induced a significant increase in the inflammatory cell counts. The majority of recruited cells were neutrophils and macrophages. YPL-001 treatment significantly reduced the number of neutrophils, but not macrophages, in the BALF (Figure 6D). CSE treatment induced protein leakage into BALF, which was suppressed by YPL-001 treatment (Figure 6E). We examined the effects of YPL-001 on CSE-induced expression of inflammatory chemokines and cytokines such as KC, MIP2, and IL-6 in total lung tissues of mice. The expression levels of inflammatory chemokines/cytokines were increased in total lung tissues of CSE-treated mice, and were reduced by YPL-001 treatment (Figure 6F). Moreover, we observed that CSE treatment increased the number of apoptotic cells in lung tissue sections, which was significantly suppressed by YPL-001 treatment. These data suggest that YPL-001 protects mice against CSE-induced emphysema via various mechanisms.

## 4. Discussion

CS exposure is thought to be a major risk factor in the development of COPD. Airway epithelial cells are the initial targets of CS. When airway epithelial cells are exposed to CS, the expression of inflammatory mediators increases significantly. Released pro-inflammatory mediators recruit and stimulate several types of immune cells including neutrophils, macrophages, and T lymphocytes, which induce a pulmonary inflammatory response, mucus hypersecretion, and airway remodeling [26]. Thus, suppression of the CS-induced inflammatory response in pulmonary epithelial cells may be effective in the treatment of COPD. In this study, we found that YPL-001 inhibited CSE-induced IL8 production in lung epithelial cells (BEAS-2B and primary HBECs). Similarly, it has been reported that *Veronica officinalis* extracts that include verproside (which is a component of YPL-001) inhibit TNF-α-induced pro-inflammatory mediators such as IL8 and IL6 [27]. We also found that YPL-001 completely blocked IL17A-induced IL8 release, which indicates that the inhibitory effects of YPL-001 on IL8 production are not stimulus-specific.

CSE-induced cytokine production is regulated by molecular pathways including mitogen-activated protein kinase (MAPK) and NF-κB signaling pathways [6,7]. To ascertain which pathway is involved in the YPL-001-dependent downregulation of the IL8 gene and protein expression, we determined the expression levels of active phosphorylated forms of p38, c-Jun N-terminal kinase (JNK), and extracellular signal-regulated kinase (ERK). YPL-001 pretreatment did not block the CSE-induced activation of MAPK, but rather slightly increased the level of active JNK and ERK in CSE-treated cells (data not shown), which suggests that the inhibitory effect of YPL-001 on IL8 production is independent of the MAPK pathway. Interestingly, Veronica extract also had no impact on the MAPK signaling pathway [27]. Although YPL-001 did not block the nuclear translocation of the NF-κB p65 subunit, it markedly suppressed the CSE- and TNF-α-induced transcriptional activity of NF-κB. Consistent with our data, it has been reported that verproside reduces the TNF-α-induced activity of NF-κB, resulting in suppression of TNF-α-induced *MUC5AC* expression in pulmonary mucoepidermoid carcinoma cells [28]. However, in mucoepidermoid carcinoma cells, verproside negatively regulates the upstream signaling mediators of NF-κB in the cytoplasm such as transforming growth factor beta-activated kinase 1 (TAK1), IKKα/β, and IκBα [28]. Moreover, Veronica extract, which includes verproside, inhibits NF-κB–DNA binding [27]. These results suggest that YPL-001 and its active component verproside suppress the NF-κB signaling pathway to downregulate pro-inflammatory mediators.

How does YPL-001 block NF-κB activity? The transcriptional activity of NF-κB is determined by acetylation of NF-κB subunit p65, which is regulated by histone acetyl transferase (HAT) and HDAC. Inhibition of HDAC activity increases both basal and TNF-induced expression of the NF-κB-regulated *CXCL8*, and overexpression of HDAC represses *CXCL8* expression [23,29]. Therefore, it seems that HDAC plays an important role in the regulation of NF-κB transcriptional activity. In this study, we demonstrated that CSE increased the acetylation of p65, and YPL-001 increased HDAC2 activity. Although this is not direct evidence, these data suggest that YPL-001-mediated suppression of NF-κB activity might be associated with an increase in HDAC2 activity.

Oxidative stress is an important feature of COPD, and lung epithelial cells contribute to the oxidant/anti-oxidant imbalance during oxidative stress [30]. In response to various stimuli, lung epithelial cells can produce increased amounts of ROS, which elevate oxidative stress and further increase the inflammatory and destructive response. However, airway epithelial cells synthesize and secrete numerous endogenous anti-oxidants to maintain oxidant/anti-oxidant homeostasis as a defense mechanism. As anti-oxidants inactivate reactive species and play an important role in the detoxification of toxic metabolites, the activation of the anti-oxidative pathway in epithelial cells is beneficial to COPD treatment. Importantly, our data demonstrate that YPL-001 activates the Nrf2 anti-oxidative pathway. YPL-001 increased Nrf2-dependent anti-oxidant genes such as NQO1 and HMOX1. Previous studies have shown that the genetic ablation of Nrf2 or HMOX1 enhances susceptibility to CSE-induced emphysema, and increases lung cell apoptosis [12,13]. We also observed that YPL-001 pretreatment attenuated CSE-induced lung epithelial cell apoptosis, which might be associated with the activation of the Nrf2 anti-oxidative pathway. Moreover, animal studies on mice showed that orally administered YPL-001 suppressed CSE-induced emphysema, neutrophilic inflammation, and apoptosis of lung epithelial cells in mice.

## 5. Conclusions

In conclusion, YPL-001 suppresses CSE-induced emphysema formation via multiple mechanisms such as anti-inflammatory, anti-oxidative, and anti-apoptotic effects. This study suggests that YPL-001 is a promising therapeutic agent for the treatment of inflammatory lung diseases including COPD. However, the action mechanism of each active compound of YPL-001 was not fully elucidated. To elucidate the mechanism, further detailed studies are required.

## Figures and Tables

**Figure 1 antioxidants-12-00015-f001:**
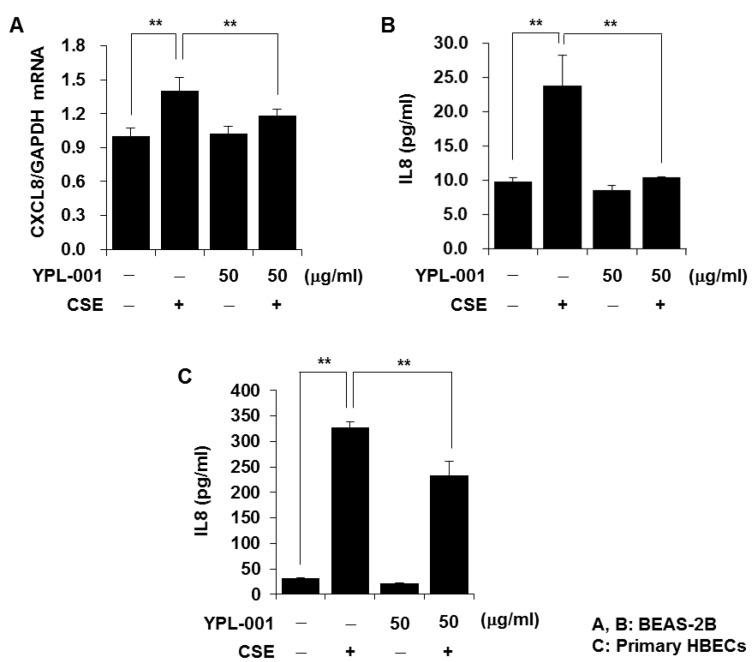
YPL-001 suppresses CSE-induced IL8 production. BEAS-2B cells were pretreated with YPL-001 for 2 h, then stimulated with CSE (1%) for 14 h (**A**) or 24 h (**B**) in the presence or absence of YPL-001. (**C**) Primary HBECs were pretreated with YPL-001 for 2 h, then stimulated with CSE (1%) for 24 h in the presence or absence of YPL-001. Quantitative real-time PCR for CXCL8 and GAPDH was performed (**A**). IL8 concentrations in the culture supernatants were determined by multiplex bead assay (**B**,**C**). Data are expressed as the mean ± SD of triplicates. ** *p* < 0.05. Results are representative of three separate experiments.

**Figure 2 antioxidants-12-00015-f002:**
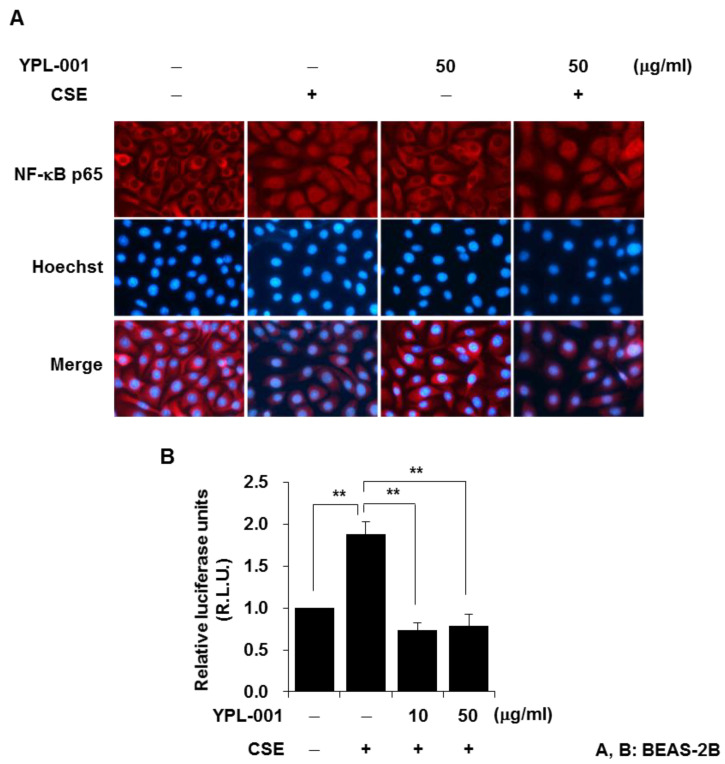
YPL-001 suppresses transcriptional activity of NF-κB. (**A**) BEAS-2B cells were pretreated with YPL-001 for 2 h, then stimulated with CSE (1%) for 1 h in the presence or absence of YPL-001. Immunofluorescent staining for p65 was performed. Hoechst dye was used as a nuclear counterstain. The cells were examined using an ECLIPSE TE300 fluorescence microscope (Nikon). (**B**) BEAS-2B cells were transiently co-transfected with the NF-κB–luciferase reporter construct and the control plasmid. Twenty-four hours after transfection, the cells were pretreated with YPL-001 for 2 h, then incubated with CSE (1%) for 8 h in the presence or absence of YPL-001. Luciferase activity was detected and normalized with renilla. Data are expressed as the mean ± SD of triplicates. ** *p* < 0.05. R.L.U. = relative luciferase units. Results are representative of three separate experiments.

**Figure 3 antioxidants-12-00015-f003:**
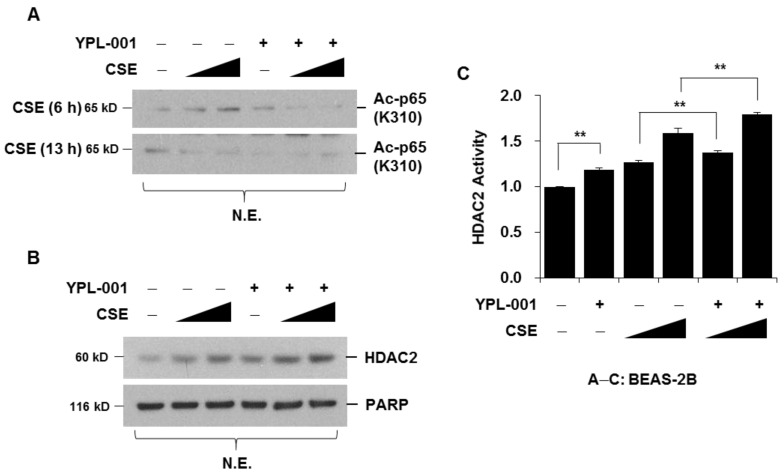
YPL-001 suppresses CSE-induced acetylation of NF-κB via increase in HDAC2 activity. (**A**) BEAS-2B cells were pretreated with YPL-001 (50 μg/mL) for 2 h, then stimulated with CSE (1, 2%) for 6 or 13 h. Nuclear extracts (N.E.) were subjected to Western blotting analysis to detect acetyl-NF-κB p65 (Lys310). (**B**,**C**) BEAS-2B cells were pretreated with YPL-001 (50 μg/mL) for 2 h, then stimulated with CSE (1, 2%) for 13 h. Nuclear extracts were subjected to Western blotting analysis to detect HDAC2 and PARP (**B**). HDAC2 activity using nuclear proteins was measured with a colorimetric assay kit (**C**). Data are expressed as the mean ± SD of triplicates. ** *p* < 0.05. Results are representative of three separate experiments.

**Figure 4 antioxidants-12-00015-f004:**
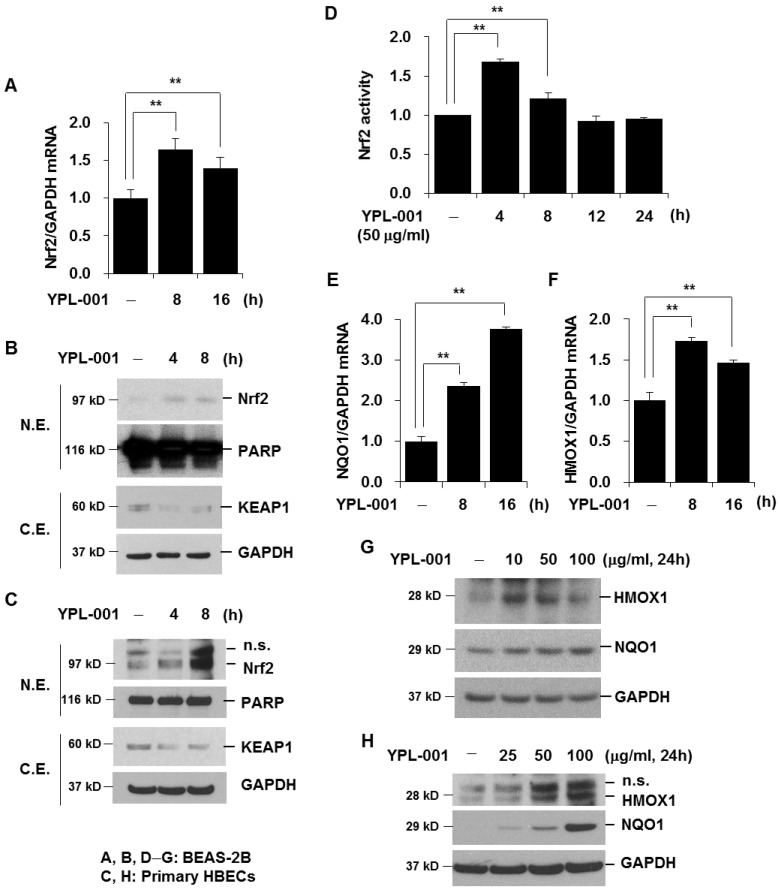
YPL-001 activates Nrf2. (**A**) BEAS-2B cells were treated with 50 μg/mL for the indicated times. Quantitative real-time PCR for Nrf2 and GAPDH was performed. Data are expressed as the mean ± SD of triplicates. ** *p* < 0.05. Both BEAS-2B cells (**B**) and primary HBECs (**C**) were treated with YPL-001 (50 μg/mL) for 4 and 8 h. Nuclear and cytoplasmic protein extracts were subjected to Western blot analysis to detect Nrf2, PARP, KEAP1, and GAPDH. (**D**) BEAS-2B cells were treated with YPL-001 for the indicated times. Nrf2 activity using nuclear proteins was measured using a DNA binding ELISA kit. Data are expressed as the mean ± SD of triplicates. ** *p* < 0.05. (**E**,**F**) BEAS-2B cells were treated with YPL-001 (50 μg/mL) for the indicated times. Quantitative real-time PCR for NQO1, HMOX1, and GAPDH was performed. Data are expressed as the mean ± SD of triplicates. ** *p* < 0.05. Both BEAS-2B cells (**G**) and primary HBECs (**H**) were treated with YPL-001 for 24 h. Total cell lysates were subjected to Western blot analysis for HMOX1, NQO1, and GAPDH. N.E. = nuclear extracts; C.E. = cytoplasmic extracts. Results are representative of three separate experiments.

**Figure 5 antioxidants-12-00015-f005:**
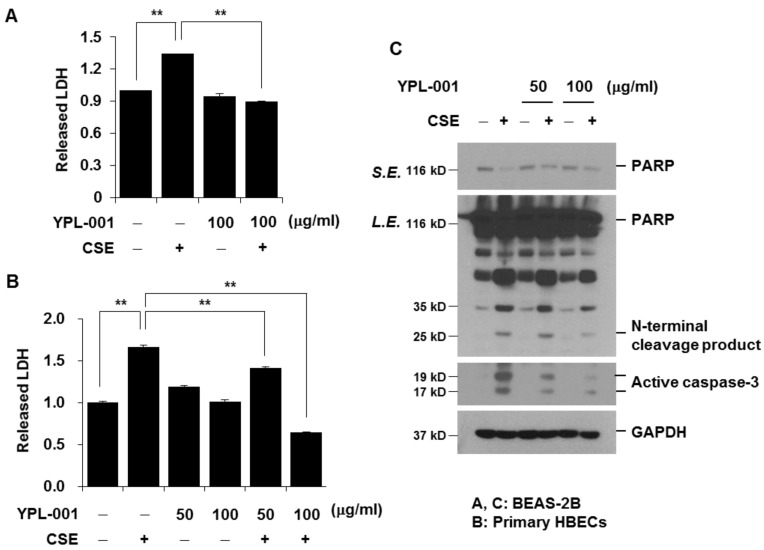
YPL-001 suppresses CSE-induced apoptotic cell death. BEAS-2B cells (**A**,**C**) and primary HBECs (**B**) were pretreated with YPL-001 for 24 h, then stimulated with CSE (4%) for 24 h in the presence or absence of YPL-001. (**A**,**B**) Cell viability was determined by LDH release assay. Data are expressed as the mean ± SD of triplicates. ** *p* < 0.05. (**C**) Total cellular extracts were subjected to Western blotting analysis to detect PARP, active caspase-3, and GAPDH. S.E. = short exposure; L.E. = long exposure. Results are representative of three separate experiments.

**Figure 6 antioxidants-12-00015-f006:**
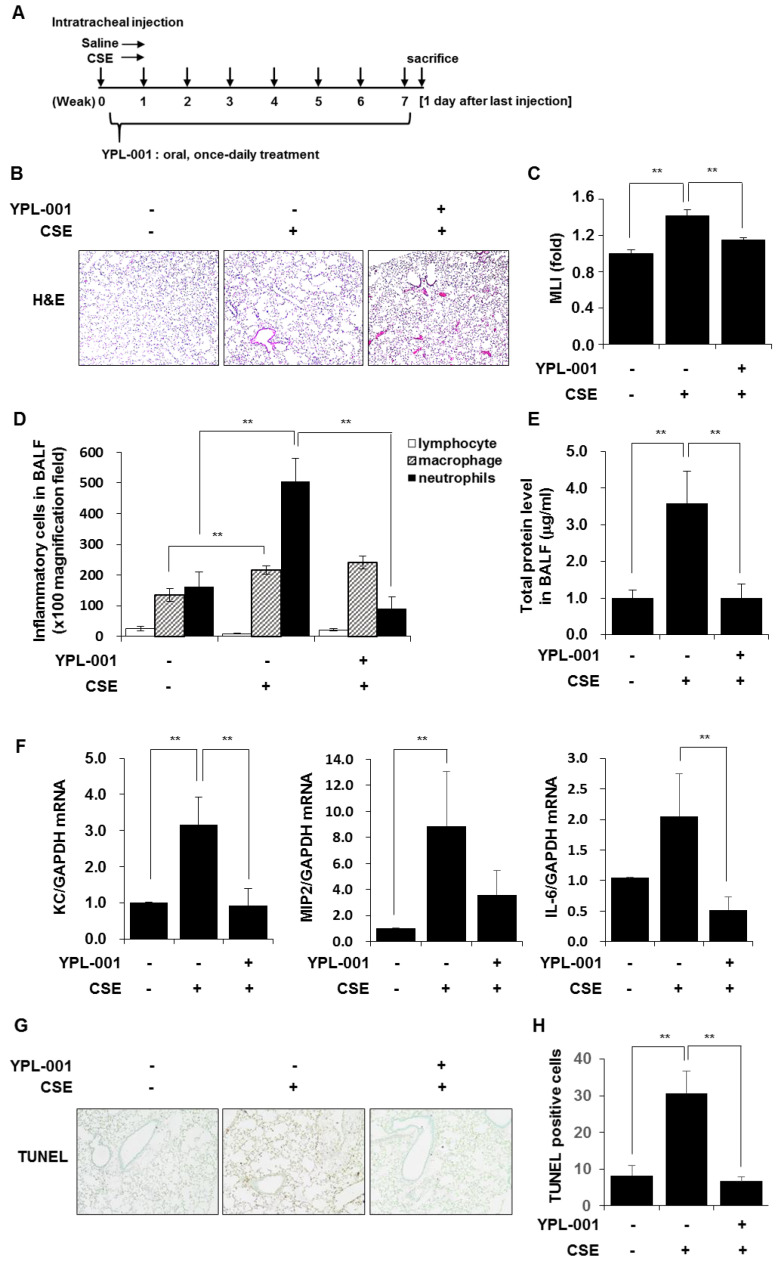
YPL-001 reduced CSE-induced emphysematous formation via multiple mechanisms such as anti-inflammatory and anti-apoptotic effects. (**A**) Experimental protocols for a murine model of emphysema. C57BL/6 mice were instilled intratracheally with saline or CSE. CSE was instilled once a week for 8 weeks (saline n = 6, CSE n = 5, YPL-001 + CSE n = 5). The mice were treated orally with YPL-001 (0.5 mg/20 g mouse) once daily for 8 weeks. The mice were sacrificed on day 1 after the last CSE instillation to collect lungs and BALF. (**B**) Representative images of H&E staining in lungs of mice (×100). (**C**) The mean linear intercept (MLI) was measured. Histogram bars represent mean ± SE. ** *p* < 0.05. (**D**) Differential cells in BALF were counted. Data are expressed as the mean ± SE. (**E**) Protein concentrations in BALF were measured. (**F**) Total RNA from the lung tissues of mice was isolated and quantitative real-time PCR for KC, MIP2, IL-6, and GAPDH was performed. Data are expressed as the mean ± SD. ** *p* < 0.05. (**G**) Representative images of TUNEL staining of lung tissue sections (×40). (**H**) Quantitation of the TUNEL-positive cells in lung tissues. Error bars represent mean ± SE. ** *p* < 0.05.

**Table 1 antioxidants-12-00015-t001:** Active compounds in YPL-001.

Active Compounds	
1	Verproside
2	Picroside II
3	6-O-veratroylcatalpol
4	Catalposide
5	Minecoside
6	Verminoside
7	Isovanillylcatalpol
8	Catalpol

## Data Availability

The data presented in this study are available within the article. Other data that support the findings of this study are available upon request to the corresponding authors.

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
