# Peer review of "YPL-001 Shows Various Beneficial Effects against Cigarette Smoke Extract-Induced Emphysema Formation: Anti-Inflammatory, Anti-Oxidative, and Anti-Apoptotic Effects"

_antioxidants, 2022, doi:10.3390/antiox12010015_

Round 1

Reviewer 1 Report

The paper presented is interesting in general well written (with only minor typo errors) and well organizing, reporting several experiments to assess the effect of YPL-001 COPD. In my opinion it deserves the publication in this journal after addressing some points reported below:

-Nuclear factor (erythroid-derived 2)-like 2 (NFE2L2) is a gene that encodes for the transcription factor Nrf2. Please stated this at row 52 and after that you must refer to Nrf2 and not NFE2L2. Please correct throughout the main text

-row 59 NFE2L2 dissociates from KEAP1 and translocates to the nucleus. Please correct in addition to NFE2L2in Nrf2, that the protein Nrf2 should be phosphorylated before entering into the nucleus.

-In the introduction, a figure reporting the molecules YPL-001 with relevant information should be added

-materials and methods the number of independent experiments should be reported for each assay.

-discussion: row 287 "We found that YPL-001 alone increased HDAC2 protein expression and HDAC activity". Authors reported that undoubtedly the compound is able to increase HDAC2 activity, but authors affirmed that increase HDAC activity. So, there are evidence that other HDAC isoforms are increased? if no authors should correct the sentence and other sentence in the main text

-due to the complex work and a lot of experiments and results obtained. I strongly recommend to add a Conclusion section summarizing all the findings in the article.

Reviewer 2 Report

Kyoung-Hee Lee et al. evaluated the effects of YPL-011 on cigarette smoke extract (CSE)-induced inflammation, on the anti-oxidative pathway and apoptosis in human lung epithelial cells and on CSE-induced emphysema in mice. The study is well designed and the results are in line with the assumptions. However, there are some changes to be made which are listed below.

Major comments

In different parts of the article there are some missing or imprecise bibliographic citations. In particular in the introduction, references 1 and 2 should be updated, if possible. Some citations are missing both in the introduction and discussion. In the introduction, for example, about CSE-induced NF-kB pathway activation. Citation 13 related to Picrorhiza scrophulariiflora. Are “Pseudolysimachion rotundum var. subintegrum” and Picrorhiza scrophulariiflora the same herb? Please specify. In addition, is citation 14 necessary to the the study? It seems not to be related to it. 

Minor comments

·         Materials and methods:

Line 194: change number of the paragraph “2.12” to “2.13”

Line 199: change number of the paragraph “2.13” to “2.14”

Line 204: change number of the paragraph “2.14” to “2.15”.

·        Results:

Figures 1A, 1B, 2B, 3A, 3B, 3C, 4A, 4B, 4D, 4E, 4F, 4G, 5A: specify cell line represented in graphics

Figure 3: data shown in Fig. 3 are related to paragraph 3.3. The authors should move Fig. 3 from paragraph 3.2 to paragraph 3.3

Line 300: review “(Fig. 5B, 5C)”

Line 311: review “(Fig. 6G, 6H)”.

Reviewer 3 Report

I really enjoyed reading this work and feel that it deserves acceptance. The introduction is good, the methods sound and the presentation of the results is excellent. This is potentially important.

The only comment I would make to improve the work would be to have a look again at the discussion. I would have liked there to be more on the mechanism of action of this compound and its potential clinical utility. It would be good to know if there are trials being done in humans and it would be good to speculate on what the pros and cons of the use of the compound may be. Also, there is little in the introduction to introduce the compound and its chemical structure which may be interesting to the readers.

I would also suggest having a lot at the figure legends which are a little hard to follow, read them through and revise.

Really well done on an excellent piece of work.
